# Novel Approaches to Environmental Monitoring and Control of *Listeria monocytogenes* in Food Production Facilities

**DOI:** 10.3390/foods11121760

**Published:** 2022-06-15

**Authors:** Priyanka Gupta, Achyut Adhikari

**Affiliations:** School of Nutrition and Food Sciences, Louisiana State University Agricultural Center, Baton Rouge, LA 70803, USA; pgupta@agcenter.lsu.edu

**Keywords:** *Listeria monocytogenes*, environmental monitoring programs, control methods, novel approaches, food production environments

## Abstract

*Listeria monocytogenes* is a serious public health hazard responsible for the foodborne illness listeriosis. *L. monocytogenes* is ubiquitous in nature and can become established in food production facilities, resulting in the contamination of a variety of food products, especially ready-to-eat foods. Effective and risk-based environmental monitoring programs and control strategies are essential to eliminate *L. monocytogenes* in food production environments. Key elements of the environmental monitoring program include (i) identifying the sources and prevalence of *L. monocytogenes* in the production environment, (ii) verifying the effectiveness of control measures to eliminate *L. monocytogenes*, and (iii) identifying the areas and activities to improve control. The design and implementation of the environmental monitoring program are complex, and several different approaches have emerged for sampling and detecting *Listeria monocytogenes* in food facilities. Traditional detection methods involve culture methods, followed by confirmation methods based on phenotypic, biochemical, and immunological characterization. These methods are laborious and time-consuming as they require at least 2 to 3 days to obtain results. Consequently, several novel detection approaches are gaining importance due to their rapidness, sensitivity, specificity, and high throughput. This paper comprehensively reviews environmental monitoring programs and novel approaches for detection based on molecular methods, immunological methods, biosensors, spectroscopic methods, microfluidic systems, and phage-based methods. Consumers have now become more interested in buying food products that are minimally processed, free of additives, shelf-stable, and have a better nutritional and sensory value. As a result, several novel control strategies have received much attention for their less adverse impact on the organoleptic properties of food and improved consumer acceptability. This paper reviews recent developments in control strategies by categorizing them into thermal, non-thermal, biocontrol, natural, and chemical methods, emphasizing the hurdle concept that involves a combination of different strategies to show synergistic impact to control *L. monocytogenes* in food production environments.

## 1. Introduction

*Listeria monocytogenes* continues to be a significant cause of foodborne illnesses. *L. monocytogenes* is a motile, facultative anaerobic, gram-positive, non-spore forming, rod-shaped bacteria which thrives between −0.4 °C to 50 °C [1]. *L. monocytogenes* was first described in 1926 by Murray et al. during investigating infected laboratory guinea pigs and rabbits [2], it was not until the 1980s that it was considered a serious public health hazard and a foodborne pathogen [3]. The bacteria occur ubiquitously in nature and has been found to be widely present in surface water, soil, plants, silage, sewage, slaughterhouse waste, and cow milk [4]. Its ability to thrive in various environmental stresses, such as low pH, high salt concentration, and low temperature, favors it as a foodborne pathogen [4].

Listeriosis is a serious foodborne illness caused by this pathogen, especially in susceptible populations, including children, pregnant women, the elderly, and individuals with compromised immune systems [5]. The symptoms of listeriosis include mild flu-like infection to severe cases of invasive infection, in which the bacteria spread from intestines to the blood, causing bloodstream infection, or central nervous system infection, causing meningitis and encephalitis [5]. In pregnant women, the infection may get transmitted from mother to neonate, causing spontaneous abortion or the birth of a premature infant with meningitis. The Centers for Disease Control and Prevention (CDC) estimates that listeriosis is the third leading cause of death from foodborne illnesses, with a fatality rate of 20 to 30% [6]. Epidemiological studies reveal that out of 13 identified serotypes (1/2a, 1/2b, 1/2c, 3a, 3b, 3c, 4a, 4ab, 4b, 4c, 4d, 4e and 7) of *L. monocytogenes*, three serotypes 1/2a, 1/2b, and 4b account for 90% of human listeriosis cases [7].

A total of 81 listeriosis outbreaks were reported by CDC’s Foodborne Disease Outbreak Surveillance System (FDOSS) from 2010 to 2021, resulting in 829 illnesses, 728 hospitalizations, and 133 deaths in the U.S. (Figure 1) [8]. Several food products were linked to these outbreaks, including packaged salad, ready-to-eat deli sliced meats, ready-to-eat pork products, refrigerated smoked fish, raw or undercooked poultry, hard-boiled eggs, cheeses, milk, ice cream, dips, prepackaged leafy greens, sprouts, and raw or processed fruits and vegetables. Ready-to-eat (RTE) foods are the common source of listeriosis, as these foods are usually consumed without additional processing by the consumer. A long shelf life makes RTE foods susceptible to bacterial growth, especially foods with intrinsic factors that support the growth of *L. monocytogenes*, e.g., foods with pH between 4.5 and 9 [9] and water activity ≥ 0.92 [10]. Studies have reported the presence of *L. monocytogenes* in RTE foods during storage at a refrigerated temperature [9]. Contamination of the food production environment plays an important role in the contamination of RTE foods, including the products that do not undergo a killing step during the production process, such as cheeses, salad, and fresh-cut produce [9,10].

Several multistate listeriosis outbreaks have been linked to RTE foods over the years. In two separate outbreaks in 2019 and 2020, a total of 22 people were infected with listeriosis due to prepackaged deli meats (also called luncheon meats) and meats sliced at deli counters at various locations [11,12]. All 22 people were hospitalized, and two deaths were reported [11,12]. In another outbreak linked to soft cheese, 30 people were infected with listeriosis across ten states over the period of 2010 to 2015 [13]. Affected people ranged in age from 1 year to 92 years, seventy percent of affected people were females, out of which six were pregnancy related [13]. Investigation of environmental samples revealed that the contaminated production environment was responsible for the outbreak [13]. Even frozen foods and foods of plant origin have been linked with listeriosis outbreaks, e.g., ice cream, caramel apple, packaged salad, and cantaloupe [14,15,16,17]. An outbreak from the consumption of ice cream was identified in 2015, in which ten people were infected, and three people died [14]. In 2016 and 2017, the U.S. Food and Drug Administration (FDA) conducted an inspection of 89 ice cream production facilities across the country and detected the presence of *L. monocytogenes* in 19 establishments, with a prevalence of 21.3% [15]. In a sporadic outbreak from August 2014 to December 2021, 17 cases of listeriosis were linked to the packaged salad [16]. Investigations detected the presence of the outbreak strain of *L. monocytogenes* on equipment used in harvesting raw iceberg lettuce, indicating that contaminated equipment used for harvesting caused the contamination of the salad [16]. In a widespread outbreak linked to cantaloupe in 2011, 147 people were infected across 28 states, leading to 33 deaths [17]. This outbreak was one of the deadliest foodborne outbreaks in U.S. history [17].

Despite the decades of efforts of policymakers and food manufacturers, *L. monocytogenes* is a challenge in food production environments. *Listeria* is widespread in the environment, and its control in food production facilities requires a better understanding of environmental monitoring and control strategies. Therefore, the objective of this paper is to review novel approaches to environmental monitoring and control of *L. monocytogenes* in food production facilities. The following sections review the characteristics and prevalence of *L. monocytogenes* in food production environments, hygienic zoning to segregate areas with a high risk of persistent *L. monocytogenes*, environmental monitoring programs, sampling considerations, and novel approaches for the detection and control of *L. monocytogenes* in food facilities.

## 2. *L. monocytogenes* in Food Production Environments

The prevalence of *L. monocytogenes* in food production environments has been identified as a cause of many listeriosis outbreaks. For example, a listeriosis outbreak in 2011 that was linked to cantaloupe was originated from the food production environment [17]. Similarly, in 2015, the listeriosis outbreak linked to ice cream was also found to have originated in the food production environment [14]. *L. monocytogenes* may enter the food production environments through different routes, such as incoming raw material, equipment, employee activity, air flow, traffic flow, soil, water, and vegetation [18]. The prevalence of *L. monocytogenes* in a food production environment depends on several factors, including the type of food, processing method, incoming raw material, the effectiveness of cleaning and sanitation protocols, the sanitary design of equipment and facilities, and employee training [18,19]. Many studies have demonstrated that some strains of *L. monocytogenes*, once entered into the food production environment, are not completely inactivated by cleaning and sanitation processes and persist for months or years in that environment [20,21]. Studies have used different molecular subtyping methods such as amplified fragment length polymorphism (AFLP), pulsed-field gel electrophoresis (PFGE), and whole genome sequence (WGS) to find persistent strains that are repeatedly isolated from a food production environment over a period of time [22,23,24,25,26,27,28,29], and some of these studies are summarized in Table 1.

Persistent *L. monocytogenes* is difficult to eliminate because it is present in a “niche” within the facility or equipment that can be difficult to clean and sanitize [21]. Such niches include cracks and crevices in different parts of the facility and equipment, such as floors, walls, drains, pipes, conveyors, mixers, slicers, freezers, condensers, gaskets, trollies, packaging machines, and so forth. These are the locations which are difficult to reach and where food particles and microorganisms tend to harbor. The persistence of *L. monocytogenes* in harborage sites depends on the efficacy of cleaning and sanitation process and the number of cells prior to and after cleaning and sanitation [25]. For example, if the reduction in the number of bacterial cells in harborage site after cleaning and sanitation is less than the increase in the number of cells due to growth, the bacterial strain persists in the harborage site. Conversely, transient strains of *L. monocytogenes* are removed with normal cleaning and sanitation and do not persist in a food production environment. Even with good cleaning and sanitation practices, transient strains may appear from time to time in an establishment and may be detected occasionally through testing.

Several scientists have questioned the relationship between the persistence of *L. monocytogenes* and its ability to form biofilms [30,31,32,33]. However, both concepts are not fully understood and require more in-depth discussion. Biofilms have been defined as the population of microbial cells adherent to each other and/or to the surfaces by producing a three-dimensional extracellular matrix [30,34,35]. The formation of biofilms occurs in sequential steps [36,37]: (1) attachment of planktonic bacteria to solid surface by electrostatic forces, van der Waals forces, and hydrophobic interactions, (2) proliferation of cells and formation of extracellular polymeric substances, (3) construction of multilayer cellular clusters with channels for the flow of nutrients and waste, and (4) biofilm formation and cell dispersion for subsequent colonization on other surfaces. *L. monocytogenes* is able to adhere to a variety of surfaces, including stainless steel, glass, propylene, rubber, quartz, marble, granite, and food surfaces such as chicken skin and beef surfaces [38,39,40]. *L. monocytogenes* in biofilms is protected from a variety of environmental factors, such as UV light, desiccation, acids, and toxic metals, and may survive antimicrobial and sanitizing agents such as iodine, chlorine, and quaternary ammonium compounds [30,41]. For example, Russo et al. (2018) found that sodium hypochlorite (200 ppm, *v*/*v*), hydrogen peroxide (2%, *v/v*), and benzalkonium chloride (200 ppm, *w*/*v*) were not able to completely eradicate established biofilms in experimental conditions [41]. The study also suggested that subminimal concentrations of antimicrobial and sanitizing compounds may encourage the growth of the resistant population of *L. monocytogenes*. Some studies have shown that persistent strains show biofilm formation [30,32,33], while other studies have found no relationship between persistence and biofilm formation [31].

*L. monocytogenes* can be separated into lineages using different genotypic and phenotypic approaches. Initially, *L. monocytogenes* isolates were distinguished into two lineages using multi locus enzyme electrophoresis (MLEE) and PFGE [42,43,44,45,46], and later three genetic lineages were defined based on the sequences of virulence genes, ribotyping, and genomic microarrays [47,48,49,50,51,52,53,54]. As per Nadon et al., 2001, lineage I includes serotypes 1/2b, 3b, 3c, and 4b, lineage II includes serotypes 1/2a, 3a, and 1/2c, and lineage III includes 4a and 4c [49]. Lineage I and lineage II isolates have been frequently isolated from food production environments, although some studies have reported a higher frequency of lineage II isolates [55,56,57,58]. One of the reasons for a higher prevalence of lineage II can be that lineage II can outcompete lineage I isolates if they are present in the same environment [59,60,61]. Lineage III isolates are rarely found in food production environments. Some studies have reported the relationship between certain lineages of *L. monocytogenes* and the ability to form biofilms [31], while some studies have reported no relationship between lineages and the ability to form biofilms [30,62,63].

## 3. Hygienic Zoning

Due to a well-recognized role of the production environment in the contamination of foods with *L. monocytogenes*, industry and regulatory agencies have increasingly emphasized implementing risk-based hygienic zoning and environmental monitoring programs. The purpose of hygienic zoning is to minimize the chances of transient strains to enter the sensitive areas in the food production facility [64]. The objective of the environmental monitoring program is to verify the effectiveness of hygienic zoning, determine the efficiency of cleaning and sanitation procedures, and identify niches and harborage sites so that corrective actions can be taken to eliminate persistent strains [64,65]. Hygienic zoning should be designed specifically for the facility, considering its unique production practices, equipment diversity, processing environment complexity, and history with past environmental pathogens [66]. For example, the need and scope of a zoning program for a RTE food facility can be very different from a flour milling facility. To identify high-risk areas in a facility, special attention should be paid to traffic flow patterns of raw material, personnel, semi-processed products, finished products, packaging material, waste, airflow, and activities taking place in the facility [20]. Attention should also be paid to the type of food product, its ability to support the growth of *L. monocytogenes*, and killing step applied during processing. For example, an RTE food product that is aseptically packaged after processing is at a lower risk than an RTE product that is exposed to the environment after processing [20]. Identification and differentiation of a food facility into hygienic zones help segregate areas where there is a potential risk of contamination by *L. monocytogenes* and implement hygienic practices to reduce the likelihood of contamination.

A color-coded map can be a useful tool to differentiate hygienic zones and facilitate a proper flow of traffic to minimize cross-contamination. Figure 2 shows an example of hygienic zone mapping in a hypothetical RTE food facility with different zone areas, such as non-food production areas, transitions areas, basic good manufacturing production (GMP) areas, and primary pathogen control areas [64]. The non-food production areas include maintenance areas, offices, waste areas, and staff facilities such as toilets and cafeterias, which should meet sanitation requirements but may not need to comply with GMP requirements. Transition areas include sites before entering the GMP areas, such as entry room, door, lockers, hallways, and changing rooms. These sites should be equipped with handwashing stations, foot foaming stations, physical barriers such as air showers, curtains, and turnstiles, and other personal equipment such as footwear and hairnets that are required to enter GMP areas. The GMP areas include raw material receiving areas, storage areas, and general food processing areas. These areas should follow GMP requirements, including personnel hygiene, sanitary operations, equipment maintenance, and standard operating procedures [67]. Separation of raw material handling equipment from those used for prepared products and a linear flow of products and traffic is necessary to prevent cross-contamination. Primary pathogen control areas include the part of the facility where cooked or prepared food is exposed to the environment, for example a packaging area where ready-to-eat food products are exposed to the environment. Stringent sanitation requirements, minimum personnel access, and dedicated equipment and tools are imperative to minimize contamination in this area.

## 4. Environmental Monitoring Program

An environmental monitoring program includes microbiological sampling of equipment, tools, surfaces, personnel, and facilities to detect pathogens of concern so that necessary actions can be taken to prevent contamination. Results of an environmental monitoring program for *L. monocytogenes* can help to determine the prevalence and sources of *L. monocytogenes* in the production environment, verify the effectiveness of *L. monocytogenes* control measures, and identify the areas and activities to improve control. A general strategy for environmental monitoring is to classify the facility into zones for sampling. Commonly, environmental sampling areas are divided into four zones: (1) zone 1 represents food-contact surfaces, such as utensils, conveyors, mixers, slicers, and even hands that come in direct contact with the food; (2) zone 2 represents areas immediately adjacent to food-contact surfaces and are sometimes referred to as indirect food-contact surfaces, such as equipment panels, bearings, and aprons; (3) zone 3 represents non-food contact surfaces that are within the production area, such as floors, walls, ceiling, pipes, and drains; and (4) zone 4 represents non-production areas of the facility, such as loading dock sites, hallways, and cafeterias. The objective of environmental monitoring is to find potential sources of contamination, and therefore, sampling should focus on high-risk areas that tend to have a higher frequency of contamination. The U.S. FDA recommends that zones 1 and 2 should be sampled more because contamination of these zones poses a higher risk to food safety [68]. The presence of *L. monocytogenes* in the samples collected from these zones indicates the possible contamination of *L. monocytogenes* in the product, which requires immediate corrective actions and a recall if necessary. Detection of *L. monocytogenes* in zones 3 and 4 indicates the presence of harborage sites and shows an early sign of contamination in the facility. Sampling these zones increases the chances of detecting a potential contamination source before it is found in the product [66].

### 4.1. Sampling Time and Frequency

The time and frequency of sampling may vary according to the objective of the environmental monitoring plan. For example, if the objective is to evaluate the effectiveness of cleaning and sanitation processes, samples should be taken immediately before and after cleaning and sanitizing, and before the start of the operations [25]. If the objective is to detect *L. monocytogenes* in niches and biofilms, samples should be taken during production, at least 3 h after the process has commenced or at the end of the production process before cleaning and sanitation [20]. A rotation system may be used to collect samples at different production shifts and days so that samples can be representative of entire production shifts. The sampling frequency may increase when focusing on high traffic areas and sites that are more likely to be a source of contamination, such as drip areas, standing water, crevices in the wall, and accumulated dirt in the hinges of equipment [68]. The sampling frequency may also be higher during the program’s initial phase or whenever there is a new intervention. For example, studies have evaluated the effectiveness of an intervention such as a new sanitation procedure, installation of new equipment, and employee training by conducting sampling immediately before and after the intervention and even after months to evaluate the long-term effect of the intervention [69,70]. The sampling frequency may be low in a situation where it is known that contamination cannot occur after the kill step (e.g., aseptic packaging or in-package cooking) or where microbial growth cannot occur till the time the product is consumed (e.g., dried food products) [7]. According to FDA regulatory requirements, environmental monitoring must be included in hazard evaluation whenever ready-to-eat foods (specified under regulation) are exposed to the environment before packaging and do not receive any kill step or control measure to minimize *L. monocytogenes* [71].

### 4.2. Sample Size

The number of samples plays an important role in determining the effectiveness of an environmental monitoring program. A few samples may weaken the program, whereas too many samples may incur a large cost and reduce resources [72]. The adequate number of samples may vary from facility to facility depending on different factors, such as production volume, number of shifts, number of processing lines, type of the food product, frequency of cleaning and sanitation processes, frequency of sampling, availability of resources, and implications of previous sample analysis data [20,66]. Not all facilities are the same, and sampling number and frequency may vary case-by-case basis. The U.S. FDA recommends that even a small ready-to-eat food facility should sample at least five food-contact surface sites and five nonfood-contact surface sites per production line, with large facilities adjusting sample size depending on the size of the facility [68]. Studies suggest that a sampling plan with nonstatistical sampling consideration can also be successful if it is aggressive and well-designed [7,66]. During the initial phase of the environmental monitoring program, monitoring baseline data should be collected so that it can provide a basis for the comparison of results [64]. Baseline data collection typically involves collecting a higher number of samples over a defined period of time to capture a snapshot of routine operations and assess the occurrence of positive samples in different zones of the facility. A reduced number and frequency of samples may be collected once a baseline is established for the ongoing program [64]. Composite sampling may be done by taking samples from multiple sites and combining them to form composite samples. Composite sampling may facilitate the analysis at a reduced cost, but it is advisable only in mature programs where positives are rare [72].

### 4.3. Sample Collection Methods

Samples should be collected aseptically by trained personnel following good hygiene and good handling practices [20]. Different tools are used to collect environmental samples, such as swabs, sponges, gauze pads, polyester cloths, and contact plates, depending on the surface type and testing method. Swabbing is a commonly used method in which a swab made up of materials such as cotton, calcium alginate, and gauge is used to transfer bacteria from the surface to the swab and then from the swab into the culture medium [73]. Cotton swabbing is suitable for detecting bacteria that are loosely attached to the surface as they may have a lower recovery if bacteria are firmly attached to the surface [74]. Generally, swab samples are used to obtain qualitative results indicating the presence or absence of bacteria and are not used for quantification [20]. Studies have shown that a sponge can retain a higher number of bacteria due to its porous structure [75]. The International Organization for Standardization (ISO) recommends stick swabs for difficult to access areas and small areas, such as 10 cm^2^, and sponges for large areas, such as 25–900 cm^2^ [76]. Dry swabs should be used if samples are collected from wet surfaces, and alternatively, wet swabs with sterile diluent (e.g., phosphate-buffered saline) should be used when collecting samples from dry surfaces. Another method involves the application of replicate organism direct agar contact (RODAC) plates, in which the raised agar surface of petri plate is applied to the sample area, and the number of colonies are counted after incubation [77]. RODAC plates can provide a mirror image of bacterial contamination in the sample area but cannot be effective if the level of contamination exceeds 100 CFU/cm^2^ in the sample area [78]. A relatively new technology, the Microbial-Vac system, uses wet-vacuum technology to spray sterile buffer solutions on the surface and simultaneously collect samples from surfaces, such as cracks and crevices [79]. It uses mechanical energy to remove firmly attached bacteria, penetrate biofilms, and collect samples from hard-to-reach areas.

### 4.4. Detection Methods

The methods for detection and identification of *L. monocytogenes* in samples should be validated by recognized national or international entities, such as ISO 11290, U.S. FDA Bacteriological Analytical Manual (BAM), AOAC official methods, and USDA-FSIS method [80,81,82]. Several studies have applied methods recommended by recognized authorities for the detection of *L. monocytogenes* in foods [83,84,85,86,87]. Traditionally, the isolation and detection of *L. monocytogenes* involve using culture methods followed by confirmation methods based on phenotypic, biochemical, and immunological characterization [88]. Culture methods generally involve a two-stage enrichment of samples followed by plating on a selective differential agar [89]. Generally, the analytical sample size for foods is 25 g and the initial level of *L. monocytogenes* is low in the sample. Enrichment step suppresses the growth of other microflora in the sample and allows the growth of *L. monocytogenes* to reach a detectable level [89]. Conventional methods provide qualitative information about the presence or absence of *L. monocytogenes*, and quantitative testing may be required if the presence of *L. monocytogenes* is detected. Quantitative estimation of *L. monocytogenes* is generally done by the most probable number (MPN) technique, which provides an estimation of the density of viable organisms in a sample [90]. Studies have used MPN in combination with polymerase chain reaction (PCR) technique for rapid and reliable quantification of *L. monocytogenes* [91,92,93]. PCR amplifies specific target DNA sequence and quantifies it by detecting fluorescent probes attached to the DNA fragment [89]. Several *L. monocytogenes* specific virulence genes have been identified and targeted for PCR detection, such as *hly* (encodes listeriolysin O), *iap* (encodes invasion-associated protein p60), and *actA* (encodes Actin assembly protein), *prfA* (encodes positive regulator factor A), and *plcB* (encodes Phospholipase C protein) [88].

Detection of *L. monocytogenes* using conventional culture-based methods is simple, easy to use, and inexpensive as compared to culture-independent methods [88]. However, conventional methods are time-consuming and laborious, as they require at least 2 to 3 days to obtain results. Numerous novel detection approaches are gaining importance due to their rapidness, sensitivity, specificity, and high throughput [88]. Table 2 summarizes some rapid and novel methods by categorizing them into molecular methods, immunological methods, biosensors, spectroscopic methods, microfluidic systems, and phage-based methods.

Molecular methods detect specific DNA or RNA sequences of target organism. Quantitative PCR or real-time PCR is a method in which nucleic acid sequence is amplified using fluorescence-detecting thermocyclers and their concentration is quantified in real-time [88]. Multiplex PCR uses several sets of primers to simultaneously amplify multiple specific target genes and provides a higher throughput as compared to conventional PCR [94]. Several novel multiplex PCR assays have been developed to detect *L. monocytogenes* [94,95,96,97]. Despite being rapid, sensitive, and specific, standard PCR-based methods have certain limitations, for example, these methods are not able to detect the viability of target microorganisms [98]. To overcome this limitation, other nucleic acid-based tests to detect viable microorganisms have been extensively explored, such as real-time nucleic acid sequence-based amplification (NASBA) and loop-mediated isothermal amplification (LAMP) [99,100]. NASBA detects viable microorganisms through the amplification of messenger RNA (mRNA) [99]. Since mRNA is present only in metabolically active cells, detection of mRNA is considered to be an indicator of viable cells [101]. LAMP utilizes a DNA polymerase with strand displacement activity and 4 to 6 primers specially designed for 6 to 8 distinct sites of the target DNA [100]. The amplification generates 10^3^ times or higher copies of target sequence in less than 60 min that can be detected using fluorescence dye. Both NASBA and LAMP occur under isothermal conditions and yield large amounts of product in a short time [99,100]. Another RNA-based method is reverse transcription-PCR (RT-PCR), which uses reverse transcriptase enzyme to convert mRNA into complementary DNA (cDNA), and then cDNA is used as a template for amplification using PCR [101].

Immunological methods are based on the interaction between antibody and antigen, whereby the strength of interaction determines the sensitivity and specificity of the method. A commonly used immunological method for rapid detection of *L. monocytogenes* is enzyme-linked immunosorbent assay (ELISA). Several studies have used ELISA for detecting *L. monocytogenes* in food samples; for example, Karamonová et al. (2004) used direct sandwich ELISA and indirect competitive ELISA for the detection of *L. monocytogenes* using polyclonal antibodies against internalin protein (InlB) protein of *L. monocytogenes* [102]. Lateral flow immunoassays such as immunochromatographic strips or dipsticks provide rapid, cheap, and simple detection of microorganisms. Wang et al. (2017) developed a lateral flow assay using monoclonal antibody labelled with gold nanoparticles against invasion-associated protein p60 of *L. monocytogenes* that could detect eight common *L. monocytogenes* serotypes, including 1/2a, 1/2b, and 4b, with a detection limit of 3.7 × 10^6^ CFU/mL [103]. Ledlod et al. (2020) developed a duplex lateral flow dipstick (DLFD) test combined with LAMP to detect *L. monocytogenes* in meat products and demonstrated it to be a highly accurate method for the detection of *L. monocytogenes* [104]. Another novel approach is immunomagnetic capture, in which immunomagnetic beads coated with *Listeria*-specific antibodies are used to isolate *Listeria* from the sample in a magnetic field [105]. The beads are plated on the medium and then plates are replicated on a plastic membrane. The membrane is further treated with conjugated antibodies that bind with *Listeria*-specific antibody and a substrate to produce a color in the presence of *Listeria* colonies.

Several ready-to-use biosensor devices have been developed to test *L. monocytogenes* in food and environmental samples. The two main components in biosensors are bioreceptors and transducers [106]. Bioreceptors recognize target analytes, which can be enzymes, nucleic acid, antibodies, and antigens, and transducers convert biological interaction into measurable signals. Different types of biosensors are commercially available in the market based on the signal measured, such as optical, piezoelectric, electrochemical, and cell-based biosensors [106]. Optical biosensors detect changes in the optical field that result from the binding of analyte with bioreceptors on the surface of the transducer. Optical biosensors can be based on reflection, refraction, fluorescence, phosphorescence, resonance, dispersion, and chemiluminescence [107]. Surface plasmon resonance (SPR) is a common optical biosensor used for detecting *L. monocytogenes*. For SPR, bioreceptors are immobilized on a thin metal surface; when an analyte binds to a bioreceptor, the refractive index of metal surface changes which results in a change in wavelength required for electron resonance [107]. Raghu and Kumar, 2020 developed a novel SPR biosensor for rapid detection of *L. monocytogenes* using wheat germ agglutinin as a bioreceptor [108]. Koubová et al., 2001 indicated that SPR could detect *Listeria* at a concentration level of 10^6^ cells/mL and its sensitivity could be comparable with that of ELISA [109]. Another type of biosensor that offers simple and real-time output are piezoelectric biosensors. Surface of piezoelectric sensor is coated with bioreceptor. When an analyte binds with a bioreceptor, it changes the mass on the crystal surface, resulting a change in resonance oscillation frequency [107]. Vaughan et al., 2001 developed a quartz crystal microbalance immunosensor which could detect 1 × 10^7^ cells/mL *L. monocytogenes* in a solution [110]. The sensor could be reused ten times without any loss in activity, which could cut down the cost [110]. Sharma and Mutharasan, 2013 demonstrated the application of a novel cantilever sensor and a commercially available antibody [111]. Detection of *L. monocytogenes* at a concentration of 10^2^ cells/mL was achieved by incorporating a third antibody binding step [111]. Electrochemical sensors use bio-electrodes to convert analyte-bioreceptor interaction into measurable electrical signal. Electrochemical sensors are sensitive and can be miniaturized to use on-site for real-time detection [107]. Electrochemical sensors can be categorized based on the signals measured, i.e., ampere, potential, or impedance [107]. Cheng et al., 2014 developed a novel electrochemical immunosensor by SAM (self-assembled monolayers)-modified gold electrodes and demonstrated its application for the detection of *L. monocytogenes* in milk samples [112].

## 5. Novel Approaches to Control of *L. monocytogenes*

Different listericidal and listeriostatic approaches can be applied to control *L. monocytogenes* in foods and food production environments. Several conventional approaches, such as pasteurization, sterilization, freezing, chilling, acidification, fermentation, drying, filtration, antimicrobial agents, and additives have been used to control *L. monocytogenes* growth in foods. However, some of these approaches are harsh and adversely impact foods’ nutritional and sensory attributes [127]. In recent years, consumers have become more interested in buying food products that are minimally processed, free of additives, shelf-stable, and have a better nutritional and sensory value [128]. In order to fulfill these requirements, several novel control strategies have emerged recently. Figure 3 shows some novel approaches to control *L. monocytogenes* in foods and food production environments, based on thermal, non-thermal, biocontrol, natural, and chemical methods. Generally, a single approach is not effective in controlling *L. monocytogenes*, and a combination of approaches is required, also known as hurdle technology. For example, irradiation can be an effective approach to controlling *L. monocytogenes* in foods. However, some cells may survive and grow even after irradiation; hence, using an antimicrobial agent after irradiation may further suppress the growth of *L. monocytogenes* [129]. This deliberate and judicious combination of control strategies to create a series of hurdles that a microorganism is not able to overcome is called hurdle technology. Several studies have evaluated different hurdle strategies to control *L. monocytogenes* in food products. For example, Upadhyay et al., 2014 evaluated a combination of four plant-derived antimicrobial compounds (carvacrol, thymol, b-resorcylic acid, and caprylic acid), along with H_2_O_2_ and high-temperature treatment to control *L. monocytogenes* in cantaloupes [130]. In another study, Espina et al., 2014 applied a combination of pulsed electric field (PEF), mild heat, and natural essential oils to inactivate *L. monocytogenes* in liquid whole egg [131]. The study found that a combination of these techniques was as effective as ultra-pasteurization for killing *L. monocytogenes* but with a less detrimental impact on the product’s sensory attributes [131]. Combining two or more approaches can produce a synergistic effect by hitting different targets that disturb the homeostasis of microorganisms [132]. Several studies have successfully applied different combinations of conventional and innovative strategies to control *L. monocytogenes* in foods and food production environments [133,134,135,136,137].

### 5.1. Thermal Methods

Conventionally, thermal processing methods such as pasteurization and sterilization are applied to control *L. monocytogenes* in foods. Direct hot air, steam, heat exchangers, and hot water baths are commonly used for the thermal processing of foods at different temperature-time combinations [138]. The temperature for pasteurization ranges from 60 to 80 °C to kill microorganisms and inactivate enzymes, whereas the temperature for sterilization is >100 °C to kill spores and spore-forming bacteria [139]. D-value is the heating time required to kill 90% of microorganisms or to reduce the microbial concentration by one log. The thermal resistance of microorganisms increases with an increase in the D value [128]. For example, the D-value of *L. monocytogenes* present in milk samples ranged from 1683.7 s to 0.7 s when milk samples were heated from 52.2 to 74.4 °C [140]. Post-package pasteurization of ready-to-eat foods is gaining recognition as a useful technique to reduce the risk of post-processing contamination of *L. monocytogenes* in food products [141,142]. Post-package pasteurization or sterilization is done by packing food in the container and heating the container using steam or hot water in a retort, such as a pressure cooker or autoclave [128]. However, not all food products are suitable for high-temperature treatments as they may reduce the organoleptic quality of the product.

Several novel interventions have emerged for precise and rapid heating of foods while maintaining their sensory and nutritional characteristics, such as microwave, radio frequency, ohmic heating, and direct steam injection. In the electromagnetic spectrum, both microwave and radio frequencies belong to non-ionizing radiation, with radio frequencies ranging from 30 to 300 MHz and microwaves ranging from 300 MHz to 300 GHz [143]. The primary mechanism involved in heating with microwave and radiofrequency is dielectric heating, which is based on the interaction of molecules with dipolar nature (e.g., water) and ionic charges in foods with electromagnetic radiation oscillating at a very high frequency [143]. The dielectric system provides non-contact, uniform, and volumetric heating of the product and has been extensively evaluated for its thermal and non-thermal antimicrobial activity. Sung and Kang, 2014 assessed the effectiveness of microwave heating for the inactivation of *L. monocytogenes* and other pathogenic microorganisms in salsa products [144]. The study found that microwave heating at 915 MHz could be an alternative to pasteurization, as it can kill microorganisms while maintaining the overall quality of the product [144]. Microbial destruction by microwave occurs by denaturation of cellular protein structure, causing rupturing of the cell membrane [128]. Awuah et al., 2005 evaluated the application of radiofrequency in inactivating *Listeria* in milk and found up to 5-log reduction at 1200 W, 65 °C, and 55.5 s [145]. Radiofrequency and microwave have been applied to control microorganisms in products such as fruit juices, meat products, ready-to-eat products, coconut water, catfish, eggs, and pasta products.

Another heating method is ohmic heating or joule heating, in which foods are heated by passing an electric current through them. The amount of heat generated in the food product is directly related to the current flow through the food, depending on various extrinsic (e.g., temperature, voltage gradient, and frequency) and intrinsic (e.g., the composition of food) factors [146]. For example, electrical conductivity increases in the presence of water, ionic salts, and acids in the food [147]. In liquid foods, electrical conductivity increases with an increase in temperature but reduces as the pulp concentration increases [148]. In solid foods, the electrical conductivity depends on the food product’s bulk density, porosity, hardness, and structure [149]. Tian et al., 2018 comprehensively reviewed the impact of ohmic heating on microbial inactivation [150]. Pereira et al., 2020 identified that processing whey dairy beverages by ohmic heating results in a higher reduction rate of *L. monocytogenes* (2.10 log CFU/mL^−1^ min^−1^) and lower detrimental impact on sensory and nutritional parameters as compared to conventional processing (1.38 log CFU/mL^−1^ min^−1^) [151]. Direct steam injection is an ultra-high temperature (UHT) treatment method in which steam is injected directly into the fluid to obtain rapid, precise, and efficient heating. Roux et al., 2016 compared direct steam injection with ohmic heating in terms of their impact on the nutritional properties of liquid infant food formula and found that both methods had an equivalent impact [152]. Direct steam injection has been considered one of the best technologies to prevent thermal damage to milk [152]. The main disadvantages of a direct steam injection are the high cost and increased complexity. The water used for steam production should be potable and meet the grade standards and strict hygiene standards are needed to maintain boilers and steam equipment to prevent chemicals from entering the food [152].

### 5.2. Non-Thermal Methods

High-pressure processing (HPP) is a novel non-thermal method where a high pressure above 100 MPa is applied to the product using pressurized liquid such as water. HPP causes the inactivation of microorganisms through the mechanism of denaturing cell membrane, unfolding protein structure, changing cell membrane fluidity, ribosome dissociation, leakage of intracellular components, and eventually cell disruption [153]. The effectiveness of HPP in the inactivation of *L. monocytogenes* depends on parameters such as temperature, applied pressure, holding time, and properties and composition of the food matrix. For example, HPP is a more effective technique in the inactivation of *L. monocytogenes* in liquid foods than solid foods [154]. HPP may not always cause microbial inactivation and may sub-lethally damage the microbial cells, which may recover later. Therefore, combining HPP with other hurdles can effectively kill *L. monocytogenes*, for example, Nassau et al., 2017 evaluated the effectiveness of a combination of endolysin with HPP to inactivate *L. monocytogenes* in a buffer and found that a combination of two techniques could result in a 5-log reduction of *L. monocytogenes* cells [134]. The study indicated that HPP, when applied individually at 300 MPa for 1 min at 30 °C, could reduce the cell count by only 0.3 log CFU, but when applied in combination with endolysin, it could cause an effective inactivation of *L. monocytogenes* even at lower pressure levels.

Another alternative non-thermal method for controlling *L. monocytogenes* is the pulsed electric field (PEF), wherein the inactivation of microorganisms takes place by using high electric field pulses (>18 kV/cm) for a short time [128]. A high voltage pulsed electric field disrupts the bacterial cell membrane, leading to releasing intracellular components and eventually killing the microorganism. Several factors influence the inactivation kinetics of PEF, including electric field strength, pulse length, pulse number, temperature, pH, and conductivity [155]. Gómez et al., 2005 assessed the effectiveness of PEF on the inactivation of *L. monocytogenes* in media of different pH (3.5–7.0) and found that PEF was more effective at lower pH and higher electric field strengths [156]. At pH 3.5, a treatment of 28 kV/cm for 400 μs was able to reduce 6.0 Log_10_ cycles of *L. monocytogenes* cells [156]. In recent years, PEF technology has been explored for controlling *L. monocytogenes* in various food products such as milk and dairy products, juices, and soups.

Ultrasound is another emerging non-thermal technology for processing food products. High powered ultrasound with a frequency of 20 to 100 kHz is used in food production environments to kill microorganisms through a mechanism called cavitation [157]. In this mechanism, gas bubbles are formed in the liquid medium as a result of sonication, the bubbles expand until a critical point is reached where ultrasonic energy is insufficient to retain the vapor phase of the bubbles, hence the bubbles become unstable and collapse, creating shock waves that damage the bacterial cell wall. Baumann et al., 2009 evaluated the efficacy of ultrasound for the removal of *L. monocytogenes* biofilms from stainless steel chips and found that power ultrasound (20 kHz, 100% amplitude, 120 W, 60 s) was effective in reducing recoverable cells (3.8 log CFU/mL reduction) [158]. When ultrasonication was combined with ozonation (ozone concentration 0.5 ppm), there were no recoverable cells after the treatment (reduction of 7.31-log CFU/mL) [158]. Ultrasound is an effective technology in food processing and inactivating microorganisms, but it may impact the quality of food products by creating free radicals, off-flavors, and changing the composition of the food matrix.

Ionizing irradiation involves exposing food to radiation which causes ionization on interaction, such as gamma rays (^60^Co and ^137^Cs), high energy electron beam, or X-rays. Irradiation can be used for the decontamination of products such as meat, poultry, egg products, fish products, and spices [159]. A study by Bari et al., 2005 indicated that a low dose of ionizing irradiation can be effective in reducing *L. monocytogenes* on fresh vegetables without significantly changing the color, texture, taste, and appearance of the product [160]. Irradiation in frozen foods allows a higher dose level before developing off-flavor, for example, in frozen poultry, the dose level can be at least two times higher as compared to chilled poultry [161]. In a study by Velasco et al., 2015, electron beam was applied to eliminate *L. monocytogenes* from soft cheeses [162]. The study found that irradiation was able to reduce the bacterial load, but injured cells were recovered during storage. Combining irradiation with other hurdles can be more effective compared to irradiation alone. For example, Mohamed et al., 2011 found that a combination of gamma radiation and nisin could be effective in eliminating *L. monocytogenes* in meat products [163]. One advantage of irradiation is that it can be used to treat food in packages to reduce the risk of post-process contamination. An irradiation dose of 30–50 kGy is used to reduce microbial contamination of foods called as radappertization [164]. A high irradiation dose may cause discoloration of the product, and release of radiation-induced off-odors and off-flavors during storage. Therefore, it is important to carefully determine the dose level and apply the minimum possible doses to achieve desired level of control.

Ultraviolet radiation has germicidal effect and is used to eliminate microbial load on surfaces, air, water, and is now approved for microbial reduction in foods and juices [165]. UV-C light (254 nm) is absorbed by most microorganisms which leads to alteration in microbial DNA by dimer formation, limiting the ability of microorganism to multiply and grow. Adhikari et al., 2015 evaluated the effectiveness of UV light for the inactivation of *L. monocytogenes* on fruit surfaces and found a higher inactivation on fruits with a smoother surface, such as apples (1.6 log CFU/g reduction at 3.75 kJ/m^2^), as compared to fruits with a rough surface such as cantaloupe (1.0 log CFU/g reduction at 11.9 kJ/m^2^) [165]. Kim et al., 2002 observed a >5 log reduction of *L. monocytogenes* on stainless steel after treating with UV-C (500 μW/cm^2^) for 3 min [166]. Similar observations were made by Sommers et al., 2010, indicating the efficacy of UV-C light for routine decontamination of food-contact surfaces [167].

### 5.3. Biocontrol Methods

Biocontrol methods against *L. monocytogenes* include bacteriophages, bacteriocins, and competitive bacteria. Bacteriophages are viruses that infect and kill bacteria for propagation. Bacteriophages are highly specific toward their target bacteria and have no detrimental effect on non-target microbes, which is considered a vital advantage for biocontrol specificity and sensitivity [168]. Bacteriophages can undergo two types of life cycle: lytic and lysogenic. In the lytic cycle, bacteriophage attaches to the bacterial cell, introduces phage DNA into the bacterial cell, utilizes bacterial machinery to encode and assemble new phage particles, and at the end, releases progeny phage into the environment by lysing the bacterial cell. In the lysogenic cycle, the phage genome is integrated into the host DNA and replicates with bacterial DNA. The lysogenic phages (also known as temperate phages) continue to replicate with the host cell until an unfavorable condition occurs, which initiates the lytic cycle. Temperate phages are not suitable for biocontrol as they may not result in host cell death, whereas lytic phages are considered suitable for biocontrol due to their virulence. Bacteriophages are not considered a risk to humans upon consumption due to their high specificity towards host bacterial cells, and consequently, some commercially produced bacteriophages have been recommended as GRAS (generally recognized as safe) by FDA, such as ListShield^TM^ and Listex^TM^ P100 [169]. Gutiérrez et al., 2017 evaluated the effectiveness of ListShield^TM^ and Listex^TM^ P100 against *L. monocytogenes* in biofilms and Spanish dry-cured ham [168]. The study found that both products effectively removed 72-h old biofilm from stainless steel surface after four-hour treatment at 12 °C. Application of ListShield^TM^ on Spanish dry-cured ham was effective in lysing 100% of strains examined, whereas Listex^TM^ P100 was effective in lysing 64% of strains. The study suggested that these phage-based products can be useful for biocontrol of *L. monocytogenes* in food production environments. Phages can be applied to food products using different methods, such as spraying and dipping, or by using novel approaches such as immobilization on inert surfaces [170]. An alternative to using whole bacteriophages to control *L. monocytogenes* can be the application of endolysins. Endolysins are hydrolytic enzymes encoded by phage genome towards the end of lytic cycle to break the bacterial cell wall and release the progeny phages. Endolysins can be recombinantly produced and applied externally to bacterial cells without requiring bacteriophages. Ibarra-Sánchez evaluated the effectiveness of endolysin PlyP100 to control *L. monocytogenes* in Queso Fresco and compared it with nisin [171]. According to the study, PlyP100 showed bacterial reduction at varying *L. monocytogenes* inoculum levels, and showed no recovery at inoculum level of 1 log CFU/g. The endolysin was stable for 28 days and showed consistent antilisterial action. Nisin was not as effective as PlyP100 to control *L. monocytogenes*, however, a combination of the two showed a strong effect with no countable *L. monocytogenes* cells after 4 weeks of refrigeration [171].

Bacteriocins are ribosomally-synthesized antimicrobial peptides produced by the bacteria. Bacteriocins have the potential to kill target bacteria by creating pores in the cell membrane and by inhibiting cellular activities such as DNA replication and protein synthesis [172]. Nisin is an example of bacteriocin produced by *Lactococcus lactis* which is approved as a food preservative by the FDA [172]. Bacteriocins can be applied in food products using different methods, such as inoculating food with a bacteriocin-producing strain or extracting bacteriocin as a semi-purified or purified product and using it as an additive [173]. Studies have demonstrated the potential application of bacteriocin to control *L. monocytogenes* in food products, such as fish, meat, dairy products, salads, and juices [172,173,174]. Ming et al., 2006 applied bacteriocin (nisin and pediocin) powder on meat packaging material to test its effectiveness in inhibiting *L. monocytogenes* contamination in meats and found that packaging material coated with bacteriocins completely inhibited the growth of inoculated *L. monocytogenes* up to 12-week storage at 4 °C [175]. Vignolo et al., 2000 evaluated antilisterial activity of three bacteriocins produced by lactic acid bacteria, i.e., enterocin CRL35, nisin, and lactocin [176]. No listerial growth was observed when combinations of bacteriocins were used in meat products, suggesting that a combination of different bacteriocins can be effective in controlling *L. monocytogenes.*

In addition to bacteriocins, lactic acid bacteria produce a number of antimicrobial compounds, such as lactic acid, acetic acid, formic acid, phenylacetic acid, hydrogen peroxide, and reuterin against competing microorganisms. Reuterin, a non-proteinaceous water-soluble compound produced by *Lactobacillus reuteri*, is highly effective against certain gram-positive and gram-negative bacteria [177]. Arqués et al., 2004 reported the antilisterial activity of reuterin and indicated that reuterin at concentration 8 AU/mL in milk could cause complete inactivation of *L. monocytogenes* after incubation for 24 h at 37 °C [178]. A combination of reuterin and nisin acts synergistically to inactivate *L. monocytogenes* in milk [178]. Studies have evaluated the effectiveness of competitive-exclusion bacteria in controlling *L. monocytogenes*, especially in biofilms. For example, Zhao et al., 2004 evaluated a combination of two competitive-exclusion bacteria (*Lactococcus lactis* subsp. *lactis* C-1-92 and *Enterococcus durans* 152) for the inhibition of *Listeria* growth in biofilms in floor drains of a fresh poultry processing plant and found that the combination reduced *Listeria* count by 2.3–4.1 log CFU/100 cm^2^ at a wide temperature range of 3–26 °C [179]. Studies suggest that competitive-exclusion bacteria are able to survive and compete with *L. monocytogenes* in food production environments and can offer a practical and economic solution to prevent *Listeria* contamination.

### 5.4. Natural Methods

The application of natural and plant-derived antimicrobials, such as spices, herbs, essential oil, plant extract, and organic acids, is gaining attention for food preservation as an alternative to chemical preservatives. Natural antimicrobials have several benefits in addition to inhibiting microorganisms, for example, they increase flavor in the food, improve fragrance, improve medicinal value, and improve the nutritional quality of food [180]. Spices and herbs are derived from different parts of the plants, such as clove from flower bud which contains antimicrobial compound eugenol, cinnamon from bark which contains cinnamic aldehyde, turmeric from rhizome which contains curcumin, mustard from seeds which contains allyl isothiocyanate, and thyme and oregano from leaves which contain thymol and carvacrol [180,181,182]. Numerous studies have evaluated the antimicrobial activities of spices and herbs against *L. monocytogenes*. Ting and Deibel, 1991 examined 13 species against *L. monocytogenes* and found that cloves had bactericidal effect and oregano had bacteriostatic effect at 0.5% or 1% concentration at 4 °C and 24 °C [181]. When tested against meat slurry, a 1% concentration of clove or oregano did not have much inhibitory impact on *L. monocytogenes* [181]. Essential oils are aromatic, volatile oils obtained from different parts of the plants, including flowers, leaves, seeds, buds, roots, bark, woods, fruits, and peels. They are extracted by pressing and distillation or supercritical fluid extraction [180]. Antimicrobial activity of essential oils is due to the presence of different compounds such as terpenes, phenolic compounds, aldehydes, and esters, most of which are classified as GRAS. Studies have indicated that phenolic compounds in essential oils cause a change in the permeability of bacterial cell membrane, intervene in ATP (Adenosine 5′-triphosphate) formation, and disrupt proton motive force [182]. Several different essential oils have been evaluated for their effectiveness against *L. monocytogenes*. Sandasi et al., 2007 examined the effectiveness of five common essential oils (α-pinene, 1,8-cineole, (+)-limonene, linalool, and geranyl acetate) against biofilms [183]. Morshdy et al., 2021 examined essential oils (cinnamon bark oil, thyme oil, coriander oil, lavender oil, rosemary oil) against *L. monocytogenes* isolated from fresh retail chicken meat and found that cinnamon bark oil showed the highest antilisterial activity [184]. Studies have indicated that gram-positive bacteria are more sensitive to essential oils [182]. Essential oils due to their hydrophobic nature can easily pass through the cell wall of gram-positive bacteria, whereas the outer membrane of gram-negative bacteria possesses hydrophilic nature and limits the diffusion of essential oils [182]. One main disadvantage of using essential oils as antimicrobial agents is the production of strong aromas and off-flavors that can be undesirable in some food products.

Organic acids, such as lactic acid, malic acid, citric acid, and acetic acid are commonly used for preventing foods from microbial contamination. Organic acids inhibit microbial activity by inducing low pH stress, reducing enzymatic activity, and causing cell injury [180]. Pintado et al., 2009 evaluated the effectiveness of organic acids (lactic, malic, and citric acids) in combination with nisin against *L. monocytogenes* isolated from cheeses and found that the combination had a significant antilisterial activity than control (2N HCl, 3% [wt/vol] with nisin) [185]. Murphy et al., 2006 evaluated organic acids (acetic, lactic, propionic, and benzoic acids) in combination with steam surface pasteurization for the treatment of frankfurters during vacuum packaging and found that the combination could inhibit the growth of *L. monocytogenes* for 19 weeks at 4 °C storage temperature [186]. Several other combinations of organic acids with different hurdles have been evaluated for their synergistic effect against *L. monocytogenes*, such as a combination of organic acids with ultrasound for treating fresh lettuce [187], organic acids with ozone for treating enoki mushroom [188], and with essential oils for treating meat products [189]. A novel approach to increasing the effectiveness of organic acids is to incorporate them into edible coatings. For example, lactic acid and acetic acid incorporated in calcium alginate gels have been found to be effective against *L. monocytogenes* [180]. In another study, citric, acetic, lactic, and malic acid in chitosan coatings were effective against *L. monocytogenes* in refrigerated ready-to-eat shrimp products [190].

### 5.5. Chemical Agents

In food production facilities, cleaning and sanitation are important steps to eliminate microorganisms and dirt from food-contact surfaces, equipment, floors, and walls. Various chemical agents, such as chlorine, chlorine dioxide, hydrogen peroxide, quaternary ammonium compounds, ozone, nitrites, and phosphates are used for cleaning and sanitation. Aqueous chlorine is an effective agent to control microbial growth; however, its antimicrobial activity decreases in alkaline conditions and leads to the formation of toxic reaction products such as chloramines and trihalomethanes (THMs) [191]. Chlorine dioxide (ClO_2_) is used as an alternative to chlorine, as it is more potent (~2 times oxidation capacity) in killing bacteria and is not affected by alkaline conditions and organic compounds [192]. Researchers have investigated the application of ClO_2_ gas to disinfect several food products and food-contact surfaces [193,194,195]. ClO_2_ has more penetrability than aqueous ClO_2_ and has a better reach to microorganisms hidden in surface irregularities and biofilms [193]. Trinetta et al., 2013 evaluated the effectiveness of high-concentration short-time ClO_2_ for treating fresh produce and suggested it can be a useful technique for sanitizing produce in large-scale operations [196]. Luu et al., 2021 indicated that treatment with ClO_2_ gas (<5 mg/L) in gas permeable sachets could effectively reduce *L. monocytogenes* on strawberries and blueberries [197]. Results suggest that ClO_2_ gas has potential as a sanitizer for food processing; however, there can be economic and operational constraints in using this method on a large scale.

## 6. Conclusions

Contamination of the food production environment with *L. monocytogenes* has been shown to play an important role in the contamination of foods, including RTE foods that may or may not undergo a heat treatment or killing step. Strict adherence to good manufacturing practices, good hygiene practices, sanitation plans, and pest control plans are imperative to control *L. monocytogenes* in food facilities. To monitor and verify the effectiveness of control measures, an effective environmental monitoring program is essential. A robust, science-based environmental monitoring program includes several aspects, from identifying sampling zones to determining the time and frequency of sampling, establishing sampling procedures and detection methods, and implementing corrective actions. Not all food production environments are the same, and an environmental monitoring plan should be designed specifically for the facility on a case-by-case basis. The identification of hygienic zones helps to segregate high-risk areas in the facility where the possibility of contamination is higher. Hygienic zoning should be designed specifically for the facility, considering its unique production practices, equipment diversity, processing environment complexity, and history with past environmental pathogens. Special attention should be paid to high traffic flow areas, the type of food product, and its ability to support the growth of *L. monocytogenes*. For an environmental monitoring program, the facility should be categorized into different zones for sampling, and samples should be analyzed according to validated methods provided by recognized national or international entities. Several novel sampling and analytical approaches are gaining importance due to their rapidness, sensitivity, specificity, and high throughput. For example, molecular methods such as NASBA and LAMP are useful for detecting viable cells of *L. monocytogenes*, as well as determining its quantitative estimation. Several immunological methods based on the interaction between antibody and antigen have also been developed, such as ELISA, lateral flow immunoassay, and immunomagnetic capturing. Several ready-to-use test kits and biosensors have been developed for the rapid, cheap, and simple detection of *L. monocytogenes* in foods and food production environments. Different types of biosensors are commercially available in the market based on the signal measured, such as optical, piezoelectric, electrochemical, and cell-based biosensors. Optical biosensors are sensitive compared to other biosensors but are limited by high cost and less stability. Much research is needed to develop biosensors that are more sensitive, specific, and stable for quick and in situ detection of *L. monocytogenes*. Spectrometric methods such as NIR spectroscopy, Raman spectroscopy, MALDI-TOF MS, and hyperspectral imaging are robust, non-destructive, and rapid methods of detection. Microfluidics lab-on-a-chip are advanced microchips with integrated microprocessors, pumps, valves, thermocycler, and fluorescence detection modules for fully automated purification and detection of *L. monocytogenes* in foods. A variety of novel approaches to control *L. monocytogenes* in foods and food production environments have emerged, based on thermal, non-thermal, biocontrol, natural, and chemical methods. A single approach may not be effective in controlling *L. monocytogenes*, and a combination of approaches is required, also known as hurdle technology. Several studies have evaluated different hurdle strategies to control *L. monocytogenes* in food products. Some notable control strategies include high pressure processing, pulsed electric field, ultrasonication, ohmic heating, bacteriophages and bacteriocins, and natural antimicrobial agents. Despite recent advancements, much research is needed to develop innovative strategies that could control *L. monocytogenes* at the industrial scale. Consumers are interested in buying food products that are minimally processed and have better nutritional and sensory value. Biocontrol methods and natural antimicrobial agents hold great potential for controlling *L. monocytogenes* without compromising the quality of food; therefore, these strategies should be explored on an industrial scale. Future research is needed to develop novel high throughput methods that can detect and control *L. monocytogenes* in food production environments more efficiently.

## Figures and Tables

**Figure 1 foods-11-01760-f001:**
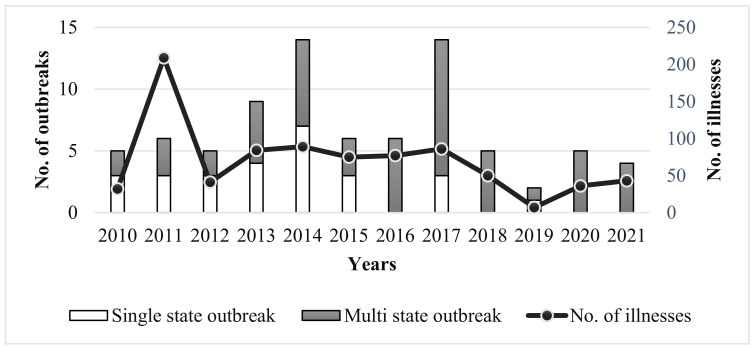
The reported single-state and multi-state outbreaks of listeriosis and total number of associated illnesses from 2010 to 2021, data derived from CDC’s Foodborne Disease Outbreak Surveillance System, United States [8].

**Figure 2 foods-11-01760-f002:**
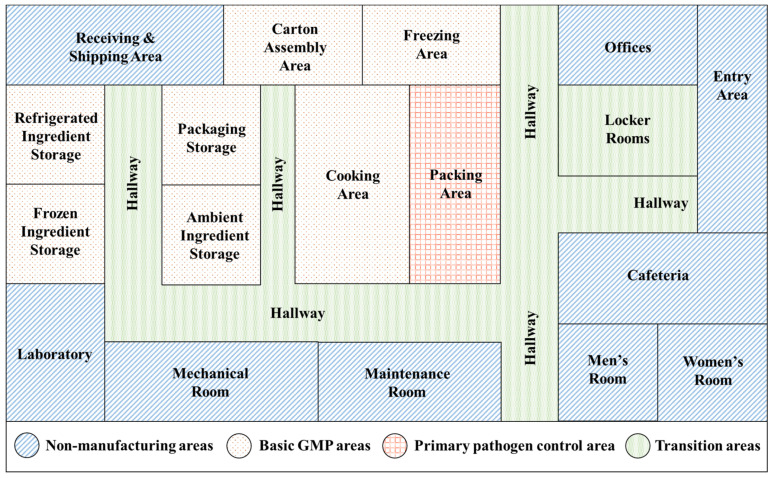
Example of hygienic zone mapping in a hypothetical ready-to-eat food facility, demonstrating different hygienic zones based on the potential risk of contamination by *L. monocytogenes*.

**Figure 3 foods-11-01760-f003:**
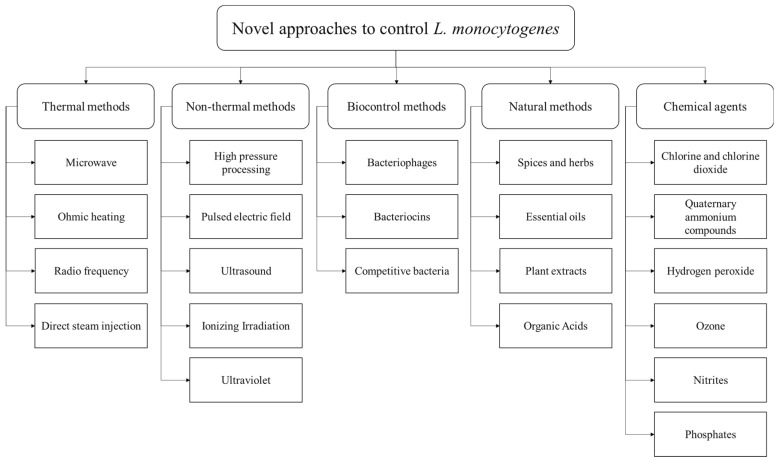
Novel approaches to control *L. monocytogenes* in food products and food production environments, categorized as thermal methods, non-thermal methods, biocontrol methods, natural methods, and chemical agents.

**Table 1 foods-11-01760-t001:** Examples of studies demonstrating that some *L. monocytogenes* strains persist in food processing environments over an extended period of time.

Food Product	Time of Persistence	Serotypes	Country	Linked to Outbreak?	References
Bulk milk	7 months	1/2a	United States	No	[23]
Cold-smoked salmon	9 months	1/2a	France	No	[24]
Goat cheese	11 months	4b	United Kingdom	Yes	[26]
Pork	1 year	Several	France	No	[27]
Salmon, seatrout, and their products	1 year	4	Poland	No	[28]
Soft cheese	5 years	ND	United States	Yes	[13]
Ice cream	7 years	1/2b	Finland	No	[29]
ND, not defined					

**Table 2 foods-11-01760-t002:** An overview of rapid and novel methods for detection of *Listeria* and *L. monocytogenes*, categorized as molecular methods, immunological methods, biosensors, spectroscopic methods, microfluidic systems, and phage-based methods.

Method	Working Principle	Advantages	Disadvantages	References
**Molecular methods**				
Multiplex PCR	Simultaneously amplifies multiple target DNA sequences and quantifies by detecting fluorescent probes attached to the DNA fragments.	Rapid and high-throughput analysis.	High cost, complex, and difficult in optimization.	[113,114]
Real-time nucleic acid sequence-based amplification (NASBA)	Amplifies nucleic acid (generally by converting single-stranded RNA into cDNA) under isothermal condition and detects fluorescent probes attached to the target fragment.	Operates without thermal cycling equipment and can detect viable microbial cells.	Complexity in handling RNA.	[99]
Loop-mediated isothermal amplification (LAMP)	Six primers target eight specific regions of target DNA, producing cauliflower-like structure of DNA bearing multiple loops. Assay performed under isothermal conditions, amplification products detected by agarose gel electrophoresis or fluorescent dye.	Greater yield, lower detection limit, operates without thermal cycling equipment.	Requires complex primer designing system, which can limit specificity.	[100]
Oligonucleotide-based microarray	A glass slide coated with chemically synthesized oligonucleotide probes detects target DNA or RNA labeled with fluorescent dye.	Simultaneous identification and typing of microbial strain.	Require high amount of target DNA or RNA.	[115]
**Immunological methods**				
Immunomagnetic capture	Labelled Immunoglobulin G and aptamer-conjugated magnetic nanoparticles form sandwich-type immuno-complex in the presence of *L. monocytogenes,* detects fluorescence.	Can detect *L. monocytogenes* without pre-enrichment.	Requires validation and further development.	[105]
Lateral flow immunoassay	Sample flows through four sections of immunoassay strip: sample pad, conjugate pad (target binds with antibody labeled by color particles), nitrocellulose pad (captures target and conjugate), and absorbent pad. Detects target as presence or absence of line colors.	Low cost, rapid, and easy to operate.	Low sensitivity and may require pre-treatment of samples.	[116]
**Biosensors**				
Optical	Detects change in optical field that results from the binding of analyte with bioreceptor on the surface of transducer. Optical biosensors can be based on reflection, refraction, fluorescence, phosphorescence, resonance, dispersion, and chemiluminescence.	Easy, rapid, and do not need pre-enrichment.	Less stability and high cost.	[107]
Piezoelectric	Surface of piezoelectric sensor is coated with bioreceptor. When analyte bind with bioreceptor it changes the mass on the crystal surface, resulting a change in resonance oscillation frequency.	Sensor can be reused.	Suitable for analytes with high molecular weight.	[117]
Electrochemical	Classified based on signal measured: ampere, potential, and impedance. Bio-electrodes are used to convert analyte-bioreceptor interaction into measurable electrical signal.	Sensitive, rapid, and cost effective.	Interference due to sample matrix.	[118]
Cell-based	Immobilized cells are used to detect analytes. Sensors or transducers are used to detect interaction between cells and analytes in terms of response time, physiological parameters, extracellular and intracellular microenvironment.	Sensitive, selective, and rapid.	Complexity in immobilizing living cells on the surface of transducers.	[119]
**Spectroscopic methods**				
Near infrared spectroscopy (NIR)	Analyzes the absorption of C-H, N-H, and O-H molecular bonds of analyte in 750–2500 nm wavelength range.	Low cost and non-destructive.	Temperature may damage samples. Interference due to water content in samples.	[120]
Raman spectroscopy	Photons of monochromatic light are absorbed and re-emitted by the sample, causing a change in the frequency of photons, called as Raman effect.	Non-destructive and high specificity.	Complex sample preparation.	[121]
Matrix-assisted laser desorption ionization time-of-flight mass spectrometry (MALDI-TOF MS)	Analyte mixed with matrix (energy absorbent organic compound) is ionized with laser beam generating protonated ions which move through a vacuum by electric field and reach a detector. The time-of-flight is detected, and mass-to-charge ratio (*m*/*z*) is measured.	Rapid, sensitive, and economical.	Limited database, low reproducibility, and limited ability to discriminate between species.	[122]
Hyperspectral imaging	Integration of conventional imaging and spectroscopy to obtain spectral and spatial information about the sample.	Non-destructive.	High limit of detection.	[123]
**Microfluidic systems**				
Microfluidics lab-on-a-chip	Microchip with integrated microprocessor, pumps, valves, thermocycler, fluorescence detection module, to purify *L. monocytogenes* cells, and detect using real time-PCR.	Fully automated purification and detection method.	Lower sensitivity.	[124]
**Phage-based methods**				
Phage protein	*Listeria* cells incubated with GFP-tagged phage protein and fluorescence measured after removal of unbound protein.	Rapid and precise glycotype determination.	Requires validation and further development.	[125]
Phage amplification	Phages replicate inside viable target cells and lyse the cells to release progeny cells along with host DNA and intracellular components which can be detected using qPCR, ELISA, or enzyme assays.	Rapid and detects viable cells.	Complex and low throughput.	[126]

## Data Availability

Data sharing is not applicable.

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
