# Peer review of "Novel Approaches to Environmental Monitoring and Control of Listeria monocytogenes in Food Production Facilities"

_foods, 2022, doi:10.3390/foods11121760_

Round 1

Reviewer 1 Report

This paper reviews the current and novel approaches to environmental monitoring and control of L. monocytogenes (Lm hereafter) in food production environments. Specifically, the paper discusses the characteristics and prevalence of Lm in food production environments, zones with a high risk of persistent Lm, sampling in environmental monitoring programs, and novel detection and control methods. Overall, the paper is comprehensive and covers an important topic, but it can be significantly improved in citations, editorial issues, and contents in some sections.

Specific points-citations

  1. In general, a significant number of references are outdated. Please consider updating them with more recent ones if possible. The journal guideline for reviewers recommends citations within the last 5 years.
  2. There are many places where citations are missing or unclear. For some content, the authors might not have added citations since they are too obvious for experts but adding citations for a broader audience may increase the completeness of this paper. In other cases, the authors mentioned citations earlier in the paragraph, but no clear connection to each statement. Here are some examples:
    • Lines 44-45, 68-70
    • Lines 70-72 (please consider adding an example)
    • Lines 77-78, 81-83, 87-89
    • Lines 143-148 (please provide references separately for each step if possible)
    • Lines 182-187
    • Lines 239-240
    • Lines 250-252 (is there a reference to provide quantitative information for this statement?)
    • Lines 289-290, 341-343, 346-349, 349-351
    • Lines 354-356, 360-367, 382-384, 415-416, 428-431, 436-439
    • Lines 602-604, 656-658, 674-676, 677-680
    • Figures 1, 2 captions

Specific points-contents

  1. Lines 99-101: 'The objective' -> 'Therefore, the objective'; Also, please add a brief guide to the following sections after this statement.
  2. Line 108: Please consider adding a brief description about where does Lm in food production environments come from.
  3. Lines 148-150: Are there other biotic (food) surfaces as well?
  4. Lines 150-153: Is this true for any concentration and treatment times?
  5. Line 170: This section should become a subsection of other sections or should be mentioned in the abstract & introduction
  6. Lines 250-253: Is there a reference to provide a quantitative guide for such an increase in sampling frequency?
  7. Line 267-268: Any quantitative information or at least the order of magnitude to differentiate 'a few' and 'too many'?
  8. Lines 462-464: Was there any time frame to report?
  9. Lines 467-472: References are too old here & probably no need to mention detailed D-values unless they are compared to novel approaches?
  10. Lines 790-792: Are there any recommendations for future directions? This could be also added at the end of each section to identify the current challenges in implementing novel approaches or other rooms for improvement.

Specific points-editorial issues

  1. Please carefully review typos, grammar, and formatting. Here are some examples:
    • Lines 3, 8, 12, 15, 31, 32: 'Monocytogenes' -> 'monocytogenes'
    • Lines 70, 97, 237, 336, 337, 600: italicize (Listeria, L. monocytogenes, prfA, plcB)
    • Lines 106, 107, 123, ...: 'Environment' -> 'Environments'
    • Line 371: caps, typo 'Enzyme-Linked Immunosorbent Assay (ELSIA)' -> 'enzyme-linked immunosorbent assay (ELISA)'
    • Line 403: 'listeria' -> 'Listeria'
    • Line 418: 'Self' -> 'self'
    • Line 694: 'el.' -> 'al.'
    • Line 739: 'et al .' -> 'et al.'
    • Table 2-Method: please use the consistent case (either title case or sentence case)
  2. Please use consistent font size. Examples are:
    • Lines 435, 437, 439, 453, 490, 495, ...
  3. Please check the units and their formatting
    • Line 413: please check if the unit '/ml' should be corrected to another cell/volume unit
    • Lines 524, 537, 553, 619, 620, 659, 660, 668: there needs to be a space between the numerical value and unit symbol
  4. Lines 40-41: It seems like this should be a continuous paragraph
  5. Line 68: 'multiply' -> 'replicate'
  6. Line 121: 'some' -> 'and some'
  7. Line 288: 'Sample collection' -> 'Sample collection method'
  8. Line 265: 'Sample number' -> 'Sample size'
  9. Line 296: 'and may have' -> 'as they may have'
  10. Line 297: 'positive or negative' -> 'qualitative'
  11. Lines 341-343: These two statements are repeated. Please consider combining them
  12. Line 346: Please consider starting a new paragraph here
  13. Line 398: 'Bioreceptors are' -> 'For SPR, bioreceptors are'
  14. Lines 401, 403: 'surface plasmon resonance' -> 'SPR'
  15. Line 726: 'food-contact' -> 'food contact' (consistency)
  16. Line 752: remove '(EMP)' as it's not referred to in the manuscript
  17. Line 753: 'determining' -> 'determine'
  18. Lines 767-768: remove 'nucleic acid sequence-based amplification' and 'Loop-mediated isothermal amplification' as their abbreviations are already defined earlier in the manuscript

Author Response

The authors would like to thank the reviewers for their time and effort in reviewing the paper.

Reviewer 1 comments and suggestions      

This paper reviews the current and novel approaches to environmental monitoring and control of L. monocytogenes (Lm hereafter) in food production environments. Specifically, the paper discusses the characteristics and prevalence of Lm in food production environments, zones with a high risk of persistent Lm, sampling in environmental monitoring programs, and novel detection and control methods. Overall, the paper is comprehensive and covers an important topic, but it can be significantly improved in citations, editorial issues, and contents in some sections.

Reviewer’s comments and suggestions      

Author’s revisions and response

Specific points-citations

In general, a significant number of references are outdated. Please consider updating them with more recent ones if possible. The journal guideline for reviewers recommends citations within the last 5 years.

Reference numbers1, 2, 4, 5, 39, 40, 42, 96, 161, 165, 166, 186, 193, 194, 199 updated with recent ones.

Some old references were removed from the manuscript.

The serial numbers of references were adjusted accordingly.

All references formatted for consistency.

There are many places where citations are missing or unclear. For some content, the authors might not have added citations since they are too obvious for experts but adding citations for a broader audience may increase the completeness of this paper. In other cases, the authors mentioned citations earlier in the paragraph, but no clear connection to each statement. Here are some examples:

Lines 44-45

Statement “Its ability to thrive in various environmental stresses, such as low pH, high salt concentration, and low temperature, favors it as a food-borne pathogen” cited with reference number 4.

Lines 68-70

Statement “L. monocytogenes can multiply in cold and wet environments of refrigerators and can survive freezers; thus, most food production environments provide ideal conditions for Listeria to thrive” removed for a better connectivity and flow of information in the paragraph.

Lines 70-72 (please consider adding an example)

Statement “Studies have reported the presence of L. monocytogenes in RTE foods during storage at a refrigerated temperature” added by citing with reference number 9.

In the statement 71- 74, examples of cheeses, fresh-cut produce, and salad were added, along with citing the statement with reference numbers 9, 10.

Lines 77-78

Statement “In another outbreak linked to soft cheese, 30 people were infected with listeriosis across ten states over the period of 2010 to 2015” cited with reference number 13.

Lines 81-83

Statement “Even frozen foods and food of plant origin have been linked with listeriosis outbreaks, e.g., ice cream, caramel apple, packaged salad, and cantaloupe” cited with reference numbers14, 16, 17. 

Lines 87-89

Statement “Frozen products do not support the growth of L. monocytogenes, but if the product becomes contaminated during the production process, the bacteria can survive in the freezer” removed to avoid redundancy.  

Lines 143-148 (please provide references separately for each step if possible)

It was not possible to add references at each step. References added at the beginning of the statement.

Lines 182-187

Statement “To identify high-risk areas in a facility, special attention should be paid to traffic flow patterns of raw material, personnel, semi-processed products, finished products, packaging material, waste, airflow, and activities taking place in the facility” cited with reference number 20.

Lines 239-240

Statement “sampling these zones increases the chances of detecting a potential contamination source before it is found in the product” cited with reference number 67.

Lines 250-252 (is there a reference to provide quantitative information for this statement?)

Statement “The sampling frequency may increase when focusing on high traffic areas and sites that are more likely to be a source of contamination, such as drip areas, standing water, crevices in the wall, and accumulated products in the hinges of equipment” cited with reference number 69.

Lines 289-290

Statement “Samples should be collected aseptically by trained personnel following good hygiene and good handling practices” cited with reference number 20.

Lines 341-343

Statement “Numerous novel detection approaches are gaining importance due to their rapidness, sensitivity, specificity, and high throughput’ cited with reference numbers 89, 99.

Lines 346-349

Statement “Quantitative PCR or real-time PCR is a method in which nucleic acid sequence is amplified using fluorescence-detecting thermocyclers and their concentration is quantified in real-time” cited with reference number 89

Lines 349-351

Statement “Multiplex PCR uses several sets of primers to simultaneously amplify multiple specific target genes and provides a higher throughput as compared to conventional PCR” cited with reference number 95.

Lines 354-356

Statement “To overcome this limitation, other nucleic acid-based tests to detect viable microorganisms have been extensively explored, such as real-time nucleic acid sequence-based amplification (NASBA) and loop-mediated isothermal amplification (LAMP)” cited with reference number 118, 119.

Lines 360-367

Statement “Another RNA-based method is reverse transcription-PCR (RT-PCR) that uses reverse transcriptase enzyme to convert mRNA into complementary DNA (cDNA), and the cDNA is used as a template for amplification using PCR” moved to the end of the paragraph and cited with reference number 99.

Lines 378- 383 cited with reference numbers 118 and 119.

Lines 382-384

Statement “Another novel approach is immunomagnetic capture, in which immunomagnetic beads coated with Listeria specific antibodies are used to isolate Listeria from the sample in a magnetic field” cited with reference number 116.

Lines 415-416

Statement “Electrochemical sensors are sensitive and can be miniaturized to use on-site for real time detection” cited with reference number 105.

Lines 428-431

Statement “However, some of these approaches are harsh and adversely impact foods’ nutritional and sensory attributes” cited with reference number 129.

Statement “In recent years, consumers have become more interested in buying food products that are minimally processed, free of additives, shelf-stable, and have a better nutritional and sensory value” cited with reference number 130.

Lines 436-439

Statement “However, some cells may survive and grow even after irradiation; hence using an antimicrobial agent after irradiation may further suppress the growth of L. monocytogenes” cited with reference number 130.

Lines 602-604

Statement “Bacteriophages are highly specific towards their target bacteria and have no detrimental effect on non-target microbes, which is considered a vital advantage for biocontrol specificity and sensitivity” cited with reference number 170.

Lines 656-658

Statement “Reuterin, a non-proteinaceous water-soluble compound produced by Lactobacillus reuteri, is highly effective against certain gram-positive and gram-negative bacteria” cited with reference number 179.

Lines 674-676

Statement “Natural antimicrobials have several benefits in addition to inhibiting microorganisms, for example, they increase flavor in the food, improve fragrance, improve medicinal value, and improve nutritional quality of food” cited with reference number 182.

Lines 677-680

Statement 697-701 cited with reference number 182.

Figures 1, 2 captions

Figure 1: Reference number 8 added in the caption.

Figure 2: Caption revised.

Specific points-contents

Lines 99-101: 'The objective' -> 'Therefore, the objective'; Also, please add a brief guide to the following sections after this statement.

Revised as suggested by the reviewer.

A brief guide to following section added.

Line 108: Please consider adding a brief description about where does Lm in food production environments come from.

A brief description (lines 118-120) added.

Lines 148-150: Are there other biotic (food) surfaces as well?

Food surfaces such as chicken skin and beef surfaces added in the statement.

Lines 150-153: Is this true for any concentration and treatment times?

Higher concentration and treatment time may have detrimental impact on L. monocytogenes in biofilms.

Line 170: This section should become a subsection of other sections or should be mentioned in the abstract & introduction

Hygienic zoning was mentioned in the introduction section.

Lines 250-253: Is there a reference to provide a quantitative guide for such an increase in sampling frequency?

Statement cited with reference number 69 (FDA-CFSAN guide).

Line 267-268: Any quantitative information or at least the order of magnitude to differentiate 'a few' and 'too many'?

Magnitude of ‘a few’ and ‘too many’ samples depend on several factors as mentioned in the manuscript. Some studies have collected as few as 4 samples while other studies have collected as many as 30,000 samples depending on the objective of the study. FDA regulations provide some indications on the adequate number of samples to be collected, as mentioned in the manuscript. Statistically, different formulas can be applied to calculate the minimum and maximum numbers of samples, which is beyond the scope of this paper.

Lines 462-464: Was there any time frame to report?

The statement “The temperature for pasteurization ranges from 60 to 80 °C to kill microorganisms and inactivate enzymes, whereas the temperature for sterilization is >100 °C to kill spores and spore-forming bacteria” focusses on the temperature range for conventional thermal treatments. Time frames are addressed in next statements by discussing D-values.

Lines 467-472: References are too old here & probably no need to mention detailed D-values unless they are compared to novel approaches?

Addressed by rephrasing the statement to give an example of D value (lines 485- 487). Example of study by Gaze et al. (1989) removed.  

Lines 790-792: Are there any recommendations for future directions? This could be also added at the end of each section to identify the current challenges in implementing novel approaches or other rooms for improvement.

Recommendations for future directions added.

Specific points-editorial issues

Please carefully review typos, grammar, and formatting. Here are some examples:

Lines 3, 8, 12, 15, 31, 32: 'Monocytogenes' -> 'monocytogenes'

'Monocytogenes' changed to 'monocytogenes' in entire manuscript.

Lines 70, 97, 237, 336, 337, 600: italicize (Listeria, L. monocytogenes, prfA, plcB)

Listeria, L. monocytogenes, prfA, plcB italicized throughout the manuscript.

Lines 106, 107, 123, ...: 'Environment' -> 'Environments'

'Environment' changed to 'Environments'.

Line 371: caps, typo 'Enzyme-Linked Immunosorbent Assay (ELSIA)' -> 'enzyme-linked immunosorbent assay (ELISA)'

'Enzyme-Linked Immunosorbent Assay (ELSIA)' changed to 'enzyme-linked immunosorbent assay (ELISA)'.

Line 403: 'listeria' -> 'Listeria'

'listeria' changed to 'Listeria'

Line 418: 'Self' -> 'self'

'Self' changed to 'self'

Line 694: 'el.' -> 'al.'

'el.' changed to 'al.'

Line 739: 'et al .' -> 'et al.'

'et al .' -> 'et al.'

Table 2-Method: please use the consistent case (either title case or sentence case)

Consistent sentence case used.

Please use consistent font size. Examples are:

Lines 435, 437, 439, 453, 490, 495, ...

Consistent font size used (Palatino Linotype, 10)

For tables: Palatino Linotype, 9

Please check the units and their formatting

Line 413: please check if the unit '/ml' should be corrected to another cell/volume unit

Units and their formatting checked.

Lines 524, 537, 553, 619, 620, 659, 660, 668: there needs to be a space between the numerical value and unit symbol

Space between numerical value and unit symbol adjusted.

Lines 40-41: It seems like this should be a continuous paragraph

Paragraph adjusted.

Line 68: 'multiply' -> 'replicate'

Line 121: 'some' -> 'and some'

Revised as suggested by the reviewer.

Line 288: 'Sample collection' -> 'Sample collection method'

Revised as suggested by the reviewer.

Line 265: 'Sample number' -> 'Sample size'

Revised as suggested by the reviewer.

Line 296: 'and may have' -> 'as they may have'

Revised as suggested by the reviewer.

Line 297: 'positive or negative' -> 'qualitative'

Revised as suggested by the reviewer.

Lines 341-343: These two statements are repeated. Please consider combining them

Statement “Rapid detection methods detect L. monocytogenes in time efficient and labour-saving manner while reducing the chances of human error” removed.

Line 346: Please consider starting a new paragraph here

Revised as suggested by the reviewer.

Line 398: 'Bioreceptors are' -> 'For SPR, bioreceptors are'

Revised as suggested by the reviewer.

Lines 401, 403: 'surface plasmon resonance' -> 'SPR'

Revised as suggested by the reviewer.

Lines 726: ‘food-contact” à ‘food contact’ (consistency)

Revised as suggested by the reviewer.

Line 752: remove '(EMP)' as it's not referred to in the manuscript

Revised as suggested by the reviewer.

Line 753: 'determining' -> 'determine'

Revised as suggested by the reviewer.

Lines 767-768: remove 'nucleic acid sequence-based amplification' and 'Loop-mediated isothermal amplification' as their abbreviations are already defined earlier in the manuscript

Revised as suggested by the reviewer.

Reviewer 2 Report

This is a very interesting, well-written review on monitoring and control of Listeria monocytogenes in foods and food production environments. Several aspects of L. monocytogenes survival, persistance, identification and prevention were presented. Although similar paper has been very recently published (Listeria monocytogenes in foods—From culture identification to 
whole-genome characteristics; DOI: 10.1002/fsn3.2910), the current review will be also interested for a broad range of readers.

There are only few typing, editing errors which should be revised:

  1. L. 3: Monocytogens writhe with a lowercase.
  2. L. 40-41: Connect these two lines.
  3. L. 321: ELSIA change change to ELISA.

Author Response

The authors would like to thank the reviewers for their time and effort in reviewing the paper.

Reviewer 2 comments and suggestions

This is a very interesting, well-written review on monitoring and control of Listeria monocytogenes in foods and food production environments. Several aspects of L. monocytogenes survival, persistence, identification and prevention were presented. Although similar paper has been very recently published (Listeria monocytogenes in foods—From culture identification to whole-genome characteristics; DOI: 10.1002/fsn3.2910), the current review will be also interested for a broad range of readers. There are only few typing, editing errors which should be revised:

Reviewer’s comments and suggestions      

Author’s revisions

L. 3: Monocytogenes write with a lowercase.

Revised as suggested by the reviewer.

L. 40-41: Connect these two lines.

Revised as suggested by the reviewer.

L. 321: ELSIA change to ELISA.

Revised as suggested by the reviewer.

Round 2

Reviewer 1 Report

The authors have addressed most of the issues raised in the previous review.

One minor follow-up question,

1. Lines 156-157 of the clear version: Since higher concentration and treatment time may kill Listeria monocytogenes in biofilms, please consider adding quantitative information here if possible. 

Author Response

The authors would like to thank the reviewers for their time and effort in reviewing the paper

Reviewer’s comments and suggestions      

Author’s revisions and response

1. Lines 156-157 of the clear version: Since higher concentration and treatment time may kill Listeria monocytogenes in biofilms, please consider adding quantitative information here if possible.

Quantitative information added and revised as per reviewer’s suggestion.

Statements changed from:

“L. monocytogenes in biofilms can survive antimicrobial and sanitizing agents like iodine, chlorine, and quaternary ammonium [42], and is protected from a variety of environmental factors, such as UV light, desiccation, acids, and toxic metals [26]. Some studies have shown that persistent strains show biofilm formation [26, 28, 29], while some studies have found no relationship between persistence and biofilm formation [27].”

to:

“L. monocytogenes in biofilms is protected from a variety of environmental factors, such as UV light, desiccation, acids, and toxic metals, and may survive antimicrobial and sanitizing agents like iodine, chlorine, and quaternary ammonium compounds [26, 42]. For example, Russo et al. (2018) found that sodium hypochlorite (200 ppm, v/v), hydrogen peroxide (2%, v/v), and benzalkonium chloride (200 ppm, w/v) were not able to completely eradicate established biofilms in experimental conditions [42]. The study also suggested that subminimal concentrations of antimicrobial and sanitizing compounds may encourage the growth of the resistant population of L. monocytogenes. Some studies have shown that persistent strains show biofilm formation [26, 28, 29], while some studies have found no relationship between persistence and biofilm formation [27].”

Few minor format edits were done.